# A Maturity Matrix Model to Strengthen the Quality Cultures in Higher Education

Niki Verschueren [1,*], Jolien Van Dessel [2], Andries Verslyppe [3], Yannick Schoensetters [1] and Martine Baelmans [4]

1    Educational Quality Monitoring Unit, KU Leuven, 3000 Leuven, Belgium
2    Faculty of Economics and Business, KU Leuven, 3000 Leuven, Belgium
3    Educational Policy Advisors, KU Leuven, 3000 Leuven, Belgium
4    Department Mechanical Engineering, KU Leuven, 3000 Leuven, Belgium
*    Correspondence: niki.verschueren@kuleuven.be

**Abstract:** This article approaches quality assurance in higher education from the perspective of quality culture. We present a concept model of quality culture that incorporates both the structural/managerial elements of the educational context as well as individual and interpersonal dynamics. The model highlights the importance of leadership, communication and information in connecting both sides of the educational praice. Our approach is unique in that it provides an interactive instrument to map, discuss and advance the existing quality cultures in cocreation with the educational actors. This instrument consists of a face-valid blueprint of the concept of quality culture. This blueprint is enriched by identifying the characteristics for less and more mature quality cultures. The feasibility of the instrument was tested in a pilot study with 13 appreciative in-depth interviews. We found that this instrument allowed faculty members and programme directors to grasp and co-create the profile of their existing quality culture. By using the appreciative approach, stimulating dialogue and reflection, our concept of quality culture aims to structure, scaffold and strengthen the continuous strive for educational quality.

**Keywords:** quality culture; higher education; quality assurance; educational quality; quality conduct; growth mind-set





## 1. Introduction

Quality assurance in higher education has long been approached as the implementation of a system to guarantee the quality of education. During the last two decades this emphasis on measuring and controlling educational output has shifted in favour of the idea of quality culture [1]. The concept of quality culture focuses on strengthening the organisational processes of the educational ecosystem in view of a continuous development of educational quality [2,3]. The concept of quality culture '[ . . . ] *refers to an organizational culture that intends to enhance quality permanently and is characterized by two distinct elements: On the one hand, a cultural/psychological element of shared values, beliefs, expectations and commitment towards quality and, on the other hand, a structural/managerial element with defined processes that enhance quality and aim at coordinating individual efforts.*' ([4], p. 10). Although this definition is elegant and face-valid, quality culture is a complex concept to put into practice. A quality culture is part of the organizational culture which by itself is a forever changing, dynamic constellation. Every level of the university—i.e., the institution, faculties, departments, programmes, didactic teams—has its own quality culture and its own contribution to the overall quality of education [5]. This multi-layered reality makes it challenging for institutions of higher education to understand the dynamics of working towards educational quality. At every level there are many factors, actors and variables at play. The quality assurance puzzle is all the more interesting given the now predominant premise that an institution of higher education should be in control of the educational

quality of its programmes [6,7]. There is an increased emphasis on the autonomy and responsibility of HEI to develop internal systems and methods of quality assurance [8].

Literature shows that every approach that focuses its quality assurance on a top-down control system in order to guarantee specific quality criteria produces negative effects, such as superficial self-assessment, unnecessary bureaucracy, increased workload, or window-dressing [9–11]. Systems are not necessarily bad, but in themselves they are insufficient to bring about a strong drive for educational quality. Bollaert [12] has discussed how quality culture and systems of quality assurance relate to educational quality; educational quality is not the result or by-product of a system of quality assurance, it is the other way around. Educational quality is the result of an existing quality culture [13,14]. This relates to the early insight of Yorke [15] that quality assurance is not so much the managing *of* the educational quality, but rather managing *for* (i.e., enabling/facilitating) educational quality. The concept of quality culture integrates top-down processes with bottom-up actions and acknowledges that they jointly contribute to a common goal: to maintain and improve educational quality.

Given the impact of quality culture on educational quality, we tried to find a way to foster the different quality cultures at every level of our university (KU Leuven, Belgium). The current article describes our approach to visualise and strengthen the existing quality cultures. Central to our approach is our aim to provide an instrument that enables educational actors to understand the dynamics of their quality culture and that can empower them into taking ownership in the development of their own quality culture. In our view, three steps are necessary to foster quality culture from within. First, we need to elaborate the *concept* of quality culture so that it can function as a mental model; the concept structure should work as a mould enabling actors to gain cognitive control of the complex, layered educational reality. When this model is shared it facilitates group focus and communication. In order to establish such a *shared mental model*, the concept model should be simple, relevant and easy to navigate. Second, we need to enrich the concept structure and build it into an instrument that makes it easy to link the concept model to everyday practice. The instrument should guarantee a consistency in the *descriptions* of a quality culture, while at the same time leaving ample space to invite cocreation and self-reflection. As a third step, in order to invite further reflection and growth, the descriptions should mark differences between *more and less mature* quality cultures.

We first outline our model of quality culture, next we will elaborate on the descriptions of less and more strong quality cultures. Doing so, this article will build towards a face-valid concept model with specific descriptions that anchor on day-to-day educational practices. The overall approach will then be tested in a pilot study. Our aim is to enable educational actors to understand, cocreate and appreciate their current quality culture, while at the same time inspiring their further development and growth.

## 2. Concept Model of Quality Culture

The concept of quality culture has been discussed by several authors [1,3,5,8,10]. In line with the definition of EUA [4], these discussions all include structural, managerial elements such as policy, resources, regulations, as well as more interactive, human factors such as shared values, engagement and trust. Every model has its own emphasis and strengths. We reviewed the different approaches in order to find a simple, face-valid mental model of quality culture. Empirical studies that shed light on the correlations and pathways between model components were taken into account [1,8,16] but found to be less suited to function as scaffold for cocreation. We decided to build from the model of quality culture by Sattler and Sonntag [17–19]. This model is directly related to the model of the EUA [4]. It has an elegant, face-valid structure and the overall logic as well as the idea behind the different dimensions can be easily explained and understood. At the same time the model can incorporate the many aspects of an educational context.

Figure 1 shows our concept map of quality culture. This conceptual model has three groups of dimensions. First, the formal–structural dimensions refer to the framework that is

in place for organising education. This includes policy objectives, organisational processes, formal flows and responsibilities, data management, infrastructure, professionalisation and tools. Second, the human dynamic dimensions refer to more intangible properties such as ownership, commitment, shared values and trust. Third, the three connecting dimensions link the formal framework to the human dynamics. These connecting dimensions are leadership, communication and participation.

**Figure 1.** Concept map of quality culture.

While the original model of Sattler and Sonntag addresses the full range of activities in higher education, including research as well as HR and overall management, we narrow the scope to education in a university setting. Additionally, while Sattler and Sonntag only define quality culture at the level of the HEI, we assign each level of the university—programme, faculty, institution—its own quality culture. At every level of a university, there is a particular way of organising the work on education. Identifying a quality culture at every level empowers actors to understand their own, specific, contribution to educational quality. It also validates the interdependent, nested organisation of university education. For instance, marking a quality culture at the level of the faculty marks the faculty's unique position to bridge the level of the study programmes to the level of the university.

The most significant difference with Sattler and Sonntag [17–19] is that we do not limit quality culture to systems of quality assurance. In our view the mapping of quality culture cannot be limited to aspects of a quality assurance system but should encompass the complex reality of every aspect of educational practice. This view relates to the idea of a regenerative quality culture, where quality assurance is inherent and indistinguishable from the everyday educational practice [10,16]. It also aligns with the approach of educational quality as a virtue of professional practice [20], the intrinsic motivation of students and professors to do well in their day-to-day teaching and learning activities. Our view also concurs with the quality culture implementation strategy for academic excellence of Freed et al. [21], who state that "*all of the systems—including leadership, development, data collection, decision making, collaboration, and planning for change—and related subsystems—such as recruitment and selection, communication, and rewards—must be designed to be congruent with each other and the supporting philosophy must be one of continuous improvement*".

## 3. Growth and Maturity in Quality Culture

Several authors agree that a strong quality culture is a necessary precondition for an educational outcome of high quality [1,12,21]. However, there is little information available on what characterises such a strong quality culture in higher education. Every

organisation exhibits their own quality culture, but this observation is not necessarily neutral or value-free. We consider some quality cultures as more mature than others in that they are better equipped to navigate towards an overall high educational quality. However, we do not consider one type of organisation as a priori more mature than another, unlike e.g., Maciag [22], who relates maturity to the principles of lean management. Rather we appreciate whatever the quality culture is and aim to empower educational actors in setting their own direction for further growth. This leaves the freedom to define growth based on the specific context. Consolidation or even deconstruction can have the same value as a drive for continuous change or lean innovation. We define more strong quality cultures by two perspectives: fit and maturity.

The first perspective to define growth is the question of overall *fit*. The quality culture arises from the functional *coherence* and interrelation between the different aspects of educational practice. The formal–structural elements are connected with human dynamics through functional leadership, communication and participation. A quality culture is functional whenever these underlying dimensions are well aligned [1,18]. This functional coherence is important, regardless of maturity. However, the specific components play out in practice, fit refers to an overall attuning of dimensions or a gut feeling to the question 'Does the organisation work?'

The second perspective of growth relates to *maturity*. Some quality cultures are considered more mature than others in that they are better suited to work towards education of high quality and even excellence. To substantiate this maturity, we define four archetypical quality cultures, ranging from less to more mature. The label 'archetype' indicates that the description fits a very typical example while the reality will be fuzzier and more differentiated. The four archetypes of quality culture development are built on different insights. First, we build from the four organisational models of quality culture by Harvey and Stensaker [10]: (1) The responsive model where quality is defined and jointly oriented towards external demands; (2) the reactive model where quality refers to compliance and is led by reward/sanction; (3) the reproductive model that favours the expertise of the individual or group, aiming at keeping status-quo; and (4) the regenerative model where there is a learning organization that focuses on internal developments, integrating external demands as added value in pursuing their own ambitions. Second, there is the extensive review by Freed et al. [21] showing that traditional, bureaucratic cultures are less effective than cultures who foster cross-functional collaboration in view of a shared ambition. Third, we integrate the five development phases for quality management of Bollaert [12]: He states that quality cultures that are less developed focus on systemic quality control and are mainly measured by quantitative data. Quality cultures that are more developed approach educational quality by use of open criteria that allow stakeholders to include whatever criteria they consider relevant. The final insight we use is that of Cheng [20] who considers an organisational culture that considers quality as a virtue of professional practice to be superior to an output-driven culture that is managed by formal objectives or consumer-oriented values. Taken together these insights allow us to carve out four archetypes of quality culture. Table 1 defines four archetypical quality cultures with increasing maturity from A to D. Quality cultures of Type A pose a manifest risk in assuring the educational quality and require immediate attention. Quality cultures that are formalistic (B) or pragmatic (C) do not necessarily present a risk in assuring educational quality. These culture types are invited to grow but remain self-regulatory. A specific context can make a less mature quality culture more adaptive, e. g., when navigating a crisis situation or in a study program with strong external regulation. When evolving towards a fully mature Type D quality culture, Type C can be considered a necessary, intermediate step. Type D represents the ideal learning organisation with a continuous, self-critical striving for improvement [15,21]. It is interesting to see that, when defining maturity in a context of lean management and business processes, the more lean/mature culture types have a stronger emphasis on human relations ([22], p. 284–286). Labelling these culture types as archetypes of increasing maturity installs a perspective of growth [23]. The maturity level

of an quality culture is considered as an emergent property [24]: It takes the maturity of the constituent dimensions into account and emerges as an overall, holistic property.

**Table 1.** Four archetypes of quality culture.

| Archetype | Quality Culture Characteristics | Reference to |
|---|---|---|
| Type A Dysfunctional us | The culture is characterised by indecisiveness, problems remain largely unresolved, the staff is only mildly engaged for educational quality, high absenteeism in meetings. | There is no overall guarantee of educational quality. There are significant risks in quality assurance [12]. |
| Type B Formalistic us←them | The culture is formal, strict and hierarchic with little flexibility. There is a minimal but sufficient guarantee of educational quality, guided by compliance with external obligations. People act based on extrinsic motivation. There is a resistance to change and a will to maintain a status-quo. | The educational quality is guaranteed at a minimal sufficient level [12]. There is a responsive or reactive quality culture [10]. There is a formal and bureaucratic organisation [20]. |
| Type C Pragmatic us→them | Some people feel compelled to work, at times they do what is needed/wanted sometimes more. The organisation has a practical orientation and works ad hoc. The organisation is aware of the opportunities of external guidelines and addresses them in a pragmatic way. | The educational quality is guaranteed and fluctuates from sufficient to good [12]. There is a reproductive quality culture [10]. Is consumer and market-oriented [20]. |
| Type D Integrated us = them | The organisation is self-critical and aims for continuous learning and growth. Strategies and external frameworks are internalised and part of the day-to-day activities. Reflections on educational quality are self-evident, it is shared as common responsibility of all stakeholders. | The educational quality is guaranteed and the continuous drive for improvement can lead to excellence [12]. The culture is regenerative [10]. There is the virtue of professional practice [20]. |

When we combine the archetypes of quality culture of Table 1 with the different dimensions of quality culture of Figure 1, we obtain a blueprint for quality culture development. The formal–structural dimensions have a more stable character [18] and the three dimensions should be aligned and generally effective. The connecting elements and the human dynamics at the group and individual level are considered to be less fixed and able to harness growth potential [17,18,21]. This concurs with Bendermacher et al. [1,8] where the main means for advancing quality cultures are elements of the connecting and group-dynamic dimensions. In our search to design an instrument that can stimulate growth, we were curious to see whether could find insights that would allow us to describe the specific properties of less and more mature quality culture types. We ran a literature review to provide a clear definition of each dimension and to investigate whether we can identify gradations in maturity. Following an abductive process, we searched within each dimension for a parallel between the growth in literature and the four archetypes of quality culture. Each dynamic dimension was taken as a separate review topic. We used the Limo database for literature review, which leverages the e-journal packages of Nature, Science, Elsevier, Wiley, JSTOR, ACM, Taylor and Francis, Springer, Cambridge, Oxford, Sage, Web of Science, etc. The initial search terms included 'quality culture' and 'higher education', combined with the respective dimensions from the model in Figure 1: leadership, communication, participation, shared values, trust, responsibility, commitment or engagement. Based on the initial return of the queries, secondary references were explored. Four researchers conducted the literature review for two or three dimensions each. They conferred regularly in order to maintain overall consistency and progress. For every dimension we examined the literature for a definition that is congruent with the concept of quality culture as described in Figure 1. We scrutinized the output for information that allowed us to assign a gradation from more to less effective or mature. Subsequently, we matched this gradation or growth path to the four archetypes of Table 1. The definitions and indications of growth for dimensions were regularly and progressively discussed in a team,

in order to manage the overall structural and conceptual cohesion. The final descriptions are the result of an abductive process combining an inductive literature review with the deductive inferences based on our priori framework of Figure 1 and Table 1. This theoretical instrument provides a general framework for strengthening quality culture but remains in service of its instrumental value. In order to obtain an intuitive instrument that sparks reflection and growth in quality culture we set face-validity and conceptual congruence as criteria. We now zoom in on each of the dimensions of the quality culture model.

## 4. Formal–Structural Elements

Educational practice is embedded in a structure of rules, agreements, roles, principles and ambitions of education. This framework is relatively stable and educational actors can readily describe, use or explain it. Every quality culture is embedded in educational practice, therefore our formal–structural framework refers to education in all its aspects. This refers to policy plans and university mission statements, the overall strategy and management model, human capital and resources as well as the overall tools and resources available [1,11,21,25–27]. When the formal framework is unclear, ill-adapted or missing, it inhibits the development of a strong quality culture [1,11,26]. The formal–structural elements of a strong quality culture are well aligned, effective and no more elaborate than is necessary. Bendermacher et al. [1] have stated that explicit consideration and scrutiny for these formal and structural aspects produce a clear improvement of the educational quality. Investigating the fitness-for-purpose of the formal–structural dimensions enables the organisation to detect incoherence or contradictions, to update out-dated processes, review regulations and cut down on needless administration. Examining and describing this framework yields specific learning effects and increases social interaction. This fosters internalisation of ambitions, strategies and resources and leads to a higher transparency of the framework for all internal stakeholders. Stakeholders gain a deeper understanding of the structures pertaining to their own everyday educational practice, as well as their relation to the other levels of the university.

While the effects of dimensions such as leadership or trust rarely stretch beyond one organisational unit or one level of the university. The formal–structural dimensions are layered. Regulations for education are carved out at the university level and trickle down to the level of student–staff interaction. As a rule, the faculties can amend or specify the university frameworks to attune to their specific disciplines, size, context or fields of study. At the level of the study programme, fundamental additions to the formal–structural framework are rare. Streamlining the formal–structural frameworks at the university or faculty level aims to be cost-efficient and can foster cross-disciplinary pollination and cooperation. Most importantly, this nesting enables the study programmes to focus on their core business of teaching and learning. A strong quality culture is aware of the larger context. They take this nesting of formal–structural dimensions into account when setting out the goals, strategies and resources at their own organisational level. At each level, the formal–structural elements refer to three dimensions: the normative, strategic and operative dimension.

### 4.1. Normative

The normative dimension refers to officially documented educational principles, goals and ambitions [17]. Its key function is to install a shared framework, a clear sense of direction and focus for all actors and stakeholders. It functions as a common compass in the striving for an education of high quality. Every level of the university has its objectives. At the level of the university, it can, e.g., include the general mission statement of the institution of higher education or frameworks such as the European Standards and Guidelines [6]. At the level of a study programme, professional qualifications can come into focus. Table 2 illustrates the three levels with some examples. For every level, the effect is clear: having a shared mission boosts morale, there is less time needed to decide how to act and people feel better about what they do [12].

In a strong quality culture these normative elements are rooted in and propagated throughout the everyday teaching and learning. The intertwining of policy and practice installs a continuous reflective loop. The vision and mission statements are regularly updated taking the insights of stakeholders into account. In a quality culture everyone is a stakeholder to be satisfied and everyone has a stakeholder to satisfy [12].

*4.2. Strategic*

The strategic elements refer to the actions and processes that contribute to the normative goals [17]. These refer to the organisational structure of the institution—the faculties, departments, research centres, study programmes—as well as to education-related procedures for assigning responsibilities and procedures for decision making. The structure and processes, the roles, responsibilities and task instructions have to be logical and coherent, clear and unambiguous [1,21]. A quality culture requires that every stakeholder knows where his responsibility lies and how she/he contributes to the objectives. A lack of clear procedures for assignment and approval as well as unclear definitions of roles and responsibilities, impede the development of a quality culture [21,27–29].

In a strong quality culture, there is a shared understanding of how the strategic layout relates to the normative goal—the quality of education—and how all actors jointly contribute. A quality culture aiming for effective quality improvement includes the anticipation of change in its strategic layout, embedding the freedom to take calculated risks and the incentives to explore new ideas and seek alternative perspectives. Freed et al. [21] consider planning for continuous change as one of the key quality principles of a higher education setting that strives for excellence.

*4.3. Operative*

The operative dimension equips the quality culture with the instruments and tools, training, and support that are needed to carry out the strategically defined steps and, ultimately, to attain the normatively defined goals [17]. In contrast, a lack of resources and instruments will hamper the growth of a quality culture [11,30]. As a rule, the faculty level makes use of the operative components set in place at the university level. They allocate resources to study programmes and invest in additional tools tailored to their specific context or discipline. Table 2 illustrates this nesting of operative elements.

Freed et al. [21] mark three components of the operative dimension. First, they refer to *technical knowledge* on education, i.e., the systematic and continuous training of educational actors, data gathering and management of information and data. Second, they refer to *resources*, including funding, materials, infrastructure and allocation of time for educational actors and stakeholders. The time that staff can allocate to teaching and educational development is one of the key determinants of a quality culture [11,30]. High workloads and lack of time are often set forward as major hurdles for taking on responsibilities in the development of a quality culture [11,30]. The availability of tools and resources is of key importance when navigating the trade-off between the roles they play in the respective domains of research, education and public debate [3,30]. Third, Freed et al. [21] refer to a system of *support*, or procedures that support approval and legitimacy.

In a strong quality culture, the operative elements provide educational actors with training for effective meetings and for fostering collaboration and participation. The information architecture allows two-way communication, consultation and the sharing of insights. Finally, the allocation of resources motivates educational actors to carve out time for their educational calling.

A question that arises is: "How can we obtain information on this formal–structural framework of a quality culture?". It is possible to give a full description of the different dimensions. However, there is a risk that by the time one finishes describing the final components, some of the descriptions are outdated. Luckily, a mature quality culture does not require a fully fleshed out and continuously updated description of the formal–structural dimensions. Because of the intricate connection between the structural educational frame-

work and the everyday educational practice, the normative goals, strategic steps and operative resources are common knowledge of educational actors. These static dimensions do not need to be fully fleshed out, rather the conceptual model of a quality culture can be used, e.g., when an update of regulations is due, when a fresh policy ambition is set out, or when a new software tool gets implemented. The model can then function as a point of reference, a means to set focus, supporting the awareness of stakeholders as to the place and impact of one formal–structural component on other components of the quality culture. In a strong quality culture stakeholders can periodically reflect on the overall formal–structural framework in order to assess whether this more static framework is still functional and agile and is no more elaborate than necessary. Such a reflection can lead to consolidation, deletion, annotation or specification of the structures and processes in place.

**Table 2.** Examples of the formal–structural dimensions on three university levels.

| Level | Normative | Strategic | Operative |
|---|---|---|---|
| University | At the level of the university the normative dimension refers to the mission statement, vision on education and learning, on quality assurance, the general policy plan and the policy plan for education. The ESG [6] are part of this normative framework and function as guidelines to inspire and assess the quality in education. | At the university level the strategic dimension refers to the flows and responsibilities as described in statutes and regulations on education. It assigns roles to educational actors and governing bodies. University wide allocation models structure financial and human resources. The governing board can set specific action plans. | At the university level the operative dimension refers to the educational processes, e.g., student administration, teaching and evaluation, the organisation of admissions, degrees, or study fees. The university also sets the financial and HR related systems, staff training and educational support, ICT, data management and teaching infrastructure. |
| Faculty | At the level of the faculty the normative dimension includes the mission statement of the faculties and their general policy plans. The faculties are relatively autonomous, building from the university policy and view on education on the one hand and on the other hand from the blueprints of the study programmes, research disciplines and student services in their care. | Every faculty has by-laws and procedures to specify the flows and responsibilities for education. The faculty decides on the overall allocation of resources and sets out agreements, decision processes and procedures that complete and tailor the regulations at the university level. | The faculty level makes use of all operative elements provided at the university level, e.g., specific training or coaching, infrastructural and financial aspects as well as specialised (information-)technology. |
| Study Programme | The blueprint: *"a concise and effective representation of the programme's rationale. The blueprint outlines what your current target for the programme is, and how you can structure the programme in relation to this ultimate objective. An essential aspect in this process is that you contextualise this rationale within the faculty and university-wide vision of education."* The dimension includes professional requirements, qualifications and disciplinary frameworks (e.g., CTI, EQUIS) | At the level of the study programme the main actors are the programme director and the Permanent Educational Committee (PEC). The PEC can structure teaching activities by grouping staff in didactic teams. Every study programme has a strategic plan that guides the study programme towards realising the ambitions of the programme. | The overall organisation of the operative elements is mainly defined at the university and faculty level. At the level of the study programme the PEC can add operative elements specific to the discipline or field of application, e.g., the practical organisation of laboratories for teaching or the organisation on internships. These add-ons are rather exceptional and limited in scope. |

## 5. Human Dynamics

The human dynamics are situated both at the collective and at the individual level [17–19]. At group level the dimensions are trust and shared values. At the level of individual ownership there is responsibility, commitment and engagement. These dimensions are also present in other approaches on quality culture e.g., [1,8,16,20,21,27]. The human dynamics are seen as actively changing properties of a quality culture model. The strongest drive for

education of high quality is to have passionate professionals in the classroom. We therefore start our outline at the individual level.

*5.1. Individual Level: Ownership*

Many authors mark ownership as an important determinant for a strong quality culture and an educational outcome of high quality [1,17,21,22,31,32]. In general, ownership generates a feeling of self-determination, vigour and purpose [22] while a lack of ownership generates a negative attitude or a minimally engaged, pragmatic attitude [1,9]. The more stakeholders are committed to contribute to the quality of education, the more they will take actions that strengthen the quality assurance [33]. Greere and Riley [34] label this dynamic a *virtuous quality cycle*.

To define growth in types of ownership, we refer to Harvey and Stensaker [10]: They state that the responsive and reactive quality cultures (Type B) are characterised by little or no sense of ownership as opposed to the reproductive and regenerative quality cultures with both a strong sense of ownership (Type C and D). We distinguish three aspects of ownership: responsibility, commitment and engagement (Figure 1). Table 3 describes the three dimensions of ownership for the four archetypes. In the literature there is no clear conceptual distinction between the four concepts. We therefore take the definitions of Sattler and Sontag [17–19] as a guideline. Table 3 provides the specific descriptions for less and for more mature archetypes of quality culture.

*Responsibility*: feeling obliged to deliver results, to fulfil the assigned role [17].
We define responsibility as the subjective obligation to care for and contribute to the overall quality of education. Responsibility is characterised by an internal sense of obligation, self-determination and critical self-judgement, taking one's own needs into account as well as the needs of others [28]. Responsibility has a cognitive component: one knows that one has to contribute based on the formal–strategic lay-out and the assigned roles. Responsibility also has an affective component: you feel responsible and obligated to commit. This affective component relates to a sense of accountability.

*Commitment*: the will to continue and carry out the task in a good way [17].
We define commitment as the degree to which members identify with the goals for education, are intrinsically motivated to continuously contribute in the best way possible and feel proud when they yield high quality results [1]. Commitment refers to a sense of duty, the degree to which actors identify with their roles. It is a necessary precondition for quality culture as well as a result of the quality culture [27]. Freed et al. [21] have stated that a quality culture exists when people are fully and equally committed to each other's success. They refer to the commitment ladder of Thompson and Roberts [35], starting from non-compliance, then to grudging compliance, formal compliance, genuine compliance, and, finally, commitment at the highest level. An effective quality culture and quality assurance requires commitment, which is facilitated by appealing the professionalism and self-improvement of academic staff [36].

*Engagement*: to act with energy and enthusiasm [17].
Engagement indicates the energy and vigour that is employed to attain the ambitions. A strong engagement translates into drive and joy, while a lack of engagement shows inactivity, aversion or indifference. We use the definition of Shaufeli and Bakker [37]: engagement is a positive, work-related mind-set that is characterised by vigour, dedication and absorption. Vigour refers to a high level of energy, mental resilience, persistence and a willingness to invest. Dedication refers to feeling inspired, a sense of significance and being enthusiastic. Absorption is experiencing flow, being immersed with a high intensity of focus.

In a strong quality culture, the three aspects of ownership come together and reinforce each other. However, individuals do not act individually, they come together in a group and group dynamics are added to the picture. For a quality culture trust and shared values are the key elements of group dynamics.

**Table 3.** Descriptions of the individual human dynamics of the quality culture model.

| | Type A | Type B | Type C | Type D |
|---|---|---|---|---|
| **Responsibility** | Responsibility is avoided, not taken on and not delegated. There is little to no accountability. One does not feel responsible. | The responsibilities are assigned in a formal and hierarchical manner. There is formal accountability. There is an extrinsic motivation guided by formal, external obligations. | Responsibility is assigned or taken on in a pragmatic way: to attain results in a fast and efficient way. At some times some members feel internally responsible/accountable for some tasks. Individual benefits and/or market share are decisive when taking on responsibility. | Responsibilities are taken on intentionally. There is staff agency for taking on responsibility/ accountability as a group. Tasks are assigned at the group level, taking into account the collective intelligence, task context and available alternatives. |
| **Commitment** | There is indifference, no personal connection. Non-compliance: *'I won't, and you can't make me. 'I will show you it will not work.'* Grudging compliance: *'I only do what I need, to keep from losing my job'.* [21], p. 130 | Compliance is focused on fulfilling the external criteria. There is little personal commitment or identification. Formal compliance: *'You said it is part of my job, so I do it.'* [21], p. 130 | Commitment is fluctuating, motivated by personal ambitions or cost/benefits. There can be high commitment to some projects or goals, depending on the temporary commitment of an individual, while other goals are not/barely met. Genuine compliance: *'It seems like a good idea to me, so tell me what you want, and I'll do it and more if I can.'* [21], p. 130 | Members are intrinsically motivated. They identify with the ambition of the organisation and stakeholders and consistently consider this a joint mission. Commitment: *'This is what I stand for. From now on everything I do will reflect this belief. I'm going to find a way to make this happen.'* [21], p. 130 |
| **Engagement** | There is indifference, absenteeism, negativity and fatigue. There is a low response rate. It is difficult to set things in motion. | There is a minimal/formal engagement. The progress is limited to prescribed actions, execution is strenuous. Tasks are fulfilled with little enthusiasm. | Engagement is varying. There is a strong, full engagement for some tasks, driven by individual enthusiasts who enjoy purpose and satisfaction. The ad hoc dedication is fragile, it can easily cool down when circumstances change. | Individuals are eager to work and feel energized. Members take pleasure in contributing to the task at hand. They get fulfilment, purpose and satisfaction from their work. |

### 5.2. Group Level: Trust

Trust implies the willingness to take risks based on the belief that the other will show reliable, honest and competent behaviour [16,38,39]. The individual who trusts is prepared to show vulnerability and take risks based on the subjective belief that the trusted entity will act competently, reliably, and honestly and will take their mutual interest and well-being into account [28,39]. Trust has been found to predict positive attitudes throughout an organisation, it implies the openness and reciprocity that lays the foundation for cohesion and cooperative actions [39,40]. It lowers the resistance to change and leads to more sustainable development [16,39]. When trust is absent or very low, the quality culture is hampered [21].

There are different approaches to trust, e.g., [1,14,16,37]. To translate the idea of maturity in quality culture to the dimension of trust, we found that the framework of Tschannen-Morran and Hoy [39] was most suitable for our approach. They distinguish five types of trust: (1) calculative trust is motivated by costly sanctions for breach of trust; (2) institution-based trust relates to the formal and informal structures of power; (3) knowledge-based trust originates from a consistency in interaction, reliability and dependability in previous situations; (4) uneven trust occurs when people trust on one level but not on another level;, finally, (5) unconditional trust begins when both parties identify

with one another, having the same goals and expectations. In the latter case there is a belief that the other party will not take advantage, will share information, ask for help, cooperate and go beyond what is expected of them. Combining these types with the insight from Freed et al. [21] which shows that calculative and institution-based trust are less effective, Table 4 identifies the types of trust for the four archetypes of quality culture. On every level we follow the advice of Lewicki et al. [41], wherein, in addition to high trust, we encourage a minimal level of distrust, inviting continuous critical thinking and open, constructive discussions and avoiding the pitfall of blind trust.

**Table 4.** Descriptions of the group level human dynamics of the quality culture model.

|  | Type A | Type B | Type C | Type D |
|---|---|---|---|---|
| **Trust** | No trust or a high level of distrust (cynical, sceptic). | Institution-based trust relates to the formal and informal structures, such as specific roles and positions of power and the overall values and norms that the organisation displays. | When people trust on one level but not on another level, there is uneven trust, e.g., personal matters versus work-related issues. Uneven trust could also refer to (sub-)group bias: given the same information, only in-group members might be given the benefit of the doubt. | Knowledge-based trust is based on the ability to predict behaviour and positive intentions. It originates from a consistency in interaction, reliability and dependability in previous situations and benefits from transparent communication. Unconditional trust: both parties identify with one another, having the same goals and expectations. There is a belief that the other party will not take advantage, will share information, ask for help and go beyond of what is expected of them. |
| **Shared values** | There are little to no shared values. | The hierarchical (bureaucratic) culture values stability and predictability [39–41]. The organisation delivers on expected results through a structured working environment, and efficient allocation of resources and smooth execution of tasks. The leaders of the organisation are efficiency-oriented, controlling organisers. Rules, procedures and policies are the primary bonding mechanisms [37]. There are clear job descriptions ensuring that the individual actions and responsibilities produce the predicted outcome. This is an internal process model (internal/control) [8] | The market-oriented culture values reputation building, goal achievement, external positioning and market superiority. The leaders set roles, goals and act as vigorous coaches. There is internal competition and external competitiveness. The adhocracy culture values flexibility, individuality and invests in a dynamic, creative, experimental working environment driven by innovation [39–41]. This is an open systems model (external/flexible), or a rational goal model (external/control) | The clan culture values a loyal, friendly working environment and invests in the long-term benefits of personal relations and human resource development. There is a focus on teamwork, participation and consensus. The organisation is characterised by flexibility, individuality, and spontaneity. The leaders of the organisations stimulate cooperation and pay attention to interpersonal cohesion and morale. Great value is attached to personal relationships, loyalty, tradition and a sense of belonging. Smart and St. John [37] have related the clan culture to the university as a community of scholars. This is a human relations model (internal/flexible) [8] |

### 5.3. Group Level: Shared Values

Shared values state what is deemed important by a group and imply a desired direction for action [36,37], they are also found to correlate to specific actions and competencies of managers [42]. Shared values play an important role in different approaches on quality

culture [27,43,44]. Tagg [45] refers to organisational values and governing habits as essential focus points in the educational development process. Bendermacher et al. [1] do not explicitly label shared values yet these are implied by their model by reference to the subculture of the organisation.

When considering growth in shared values a first perspective looks at whether the organisation has values in common. When members adhere to the same values, the work goes more smoothly [1,17]. A second perspective looks beyond mere coherence in values and checks whether some values are more effective in establishing a mature quality culture. Flexible, people-oriented cultures are generally considered to be more effective, followed by the more competitive and market-oriented cultures, while the more rigid, hierarchical cultures, aiming for predictability and stability, are considered to be less effective [5,21,43]. Freed et al. [21] assign the more mature quality culture to values such as autonomy, risk taking, collaboration, and inclusion while the less effective traditional paradigm values control, hierarchy, legitimacy and stability. To substantiate our four archetypes of quality culture we used the typology of organisational cultures of Smart and St. John [46], which in turn is based on the competing values framework (CVF) [46–48]. This framework is grounded in four models of effectiveness values, originating from two axes of competing values. Firstly, the hierarchical (bureaucratic) culture, which values stability and predictability. Rules, procedures and policies are the primary bonding mechanisms [46]. These values define an internal process model [8]. Secondly, the market-oriented culture, which values reputation building, goal achievement, external positioning and market superiority. The bonding mechanism is goal attainment and strategic, competitive action. These values define a rational goal model [8]. Thirdly, the adhocracy culture, which values flexibility, individuality and invests in a dynamic, creative, experimental working environment. The bonding mechanisms are the continuous drive for improvement, experimentation, transformational change adaptation and pioneering. These values define an open system model [8]. Finally, the clan culture, which values a loyal, friendly working environment and invests in the long-term benefits of personal relations and human resource development. The organisation is characterised by flexibility, individuality, spontaneity and a focus on cohesion and cooperation. These values define a human relation model [8]. Smart and St. John [46] found that the clan culture type scored highest on different effectiveness criteria, closely followed by the adhocracy culture type. Market cultures occupied a mid-range position, while bureaucratic culture obtained the lowest scores on all effectiveness criteria. This insight aligns with the findings of Bendermacher et al. [8] who found that the human value orientation enhanced commitment, going beyond what is expected, and has a positive effect on communication.

Overall, we can conclude that for the dimension of shared values, it is possible to identify a growth path. An organisation works more smoothly when the members share similar values. Shared values that are people-oriented are thereby considered to be more mature than formalistic or market-oriented shared values (see Table 4).

## 6. Connecting Dimensions

After describing the formal–structural dimensions (Table 2) as well as the human dynamics of a quality culture (Tables 3 and 4), it becomes clear that these relate to two very different realms of educational practice. The added value of a quality culture approach is that it connects and attunes these two realms. Following Sattler and Sonntag [17–19] we distinguish three connecting dimensions: leadership, communication and participation (Figure 1). Each of these dimensions draws on formal as well as dynamic elements. Based on our literature review we provide a working definition and explore whether there is research supporting a grading along the four archetypes of the quality culture (Table 1).

### 6.1. Leadership

EUA [4] marks leadership as one of the five conditions for attaining an effective quality culture. Indeed, leaders make the connection between the formal–structural part and the

everyday dynamics. They inspire stakeholders towards common ambitions, assign responsibilities in line with the strategy, they inform stakeholders about available tools and advocate for resources [1]. They can initiate change, confront ambiguity, install vision, stimulate cooperation and foster a climate of mutual trust and understanding [16,20,49,50]. Effective leaders act as motivator, vision setter, task master and analyser [51,52]. Leadership connects several aspects of the organisational culture. It is considered to be the binding combination of personal (charismatic, decisive), social (informal networks, goodwill), structural (systems relating to finances, HR, ICT, planning infrastructure and resources) and contextual (organisational culture, history) factors [53]. Leaders can navigate towards shared ownership by articulating a clear vision (normative) and alignment of values, by (over-)communicating on decisions and steps to be taken (strategic), by creating a resourceful and supporting environment (operative), by building trust and inviting collaboration, by being engaged, and by supporting risk taking and creativity [21].

In universities there is generally a hybrid or blended type of leadership [21,54]. The formal–structural framework defines specific roles and responsibilities (e.g., programme director, vice-president) while the academic freedom installs a shared leadership where every actor can independently or jointly contribute to the quality of education. MacBeath [55] defines four different types of leadership distribution congruent with the hybrid leadership: (1) In a formal distribution the attribution of roles and assignment of responsibilities are fully defined by formal guidelines; (2) pragmatically distributed leadership works by the cost–benefit principle, it aims for ad hoc results and is often a necessary reaction to external pressure or high workload; (3) leadership that is strategically distributed aims for a long-term goal, based on a strategically planned assignment to maximise the growth potential of the organisation; (4) incrementally distributed leadership assigns individuals more responsibility when they prove they can take on responsibility and believes in personal development and the effective delegation of tasks; (5) opportunistic leadership assigns responsibility to individuals who exhibit spontaneous initiatives and take on tasks beyond their assigned roles; finally, (6) culturally distributed leadership is assigned in an intuitive, cooperative way through engaged interaction and by maximising the collective intelligence. Leadership is collective, shared, interactive and engaged. All group members continually learn by being involved, by contributing in a constructive way and by partaking in decisions [19,53]. This leadership works when values and objectives are shared among the group members, its success depends on whether conflict can be handled in a constructive way. McBeath considers these six types as six phases in leadership development, with the cultural distribution being the most mature type [55]. This is in line with the observation that leadership styles focussing on the creation of a culture of collegiality, delegation and consultation are preferred over styles addressing quality issues through inspection and control [1,34,49]. Freed et al. [21] state that leaders should explain how each strategic decision fits with the mission and listen to stakeholders in order to improve decisions. They summarise as follows: *"Leaders are successful only when they empower others to help create and share the mission, to trust one another, to coordinate and communicate with one another, and to create and learn together"* (p. 134–135). Lomas [3] considers transformational leaders as a requirement for successful quality development, these leaders innovate and originate, focus on people rather than systems and play an active role in raising expectations. Table 5 relates the six leadership styles to the four archetypes of quality culture.

### 6.2. Communication

Communication links formal–structural goals, strategies and resources to everyday human dynamics. In a quality culture communication establishes a reciprocal and continuous dialogue between all educational actors and stakeholders [15,19]. Effective communication works in two directions. From the bottom up, it draws on ideas and input from human dynamics and facilitates interaction and participation. from the top down it informs on policies, promotes projects and resources, stimulates strategies, discusses results and encourages the use of tools and resources [1,16,21,27,51].

In a mature quality culture, there is an inclusive network of communication, where all members have access to information and have their voice heard. Instead of listening for strategic or political reasons, members listen in order to understand the context of the other and to build an inclusive network of knowledge sharing [21]. Exchanging information on ambitions, objectives and data in an open, pro-active communication promotes cooperation and synergy. When designed in this way, communication is a key factor in stimulating ownership: "*An individual without information cannot take responsibility, but an individual who is given information cannot help but take responsibility.*" ([21], p. 106). Information and communication can also harmonise different subcultures and establish collective commitment [27]. We take this as a description of highly mature quality culture, shown in Table 5, while the communication for the other quality culture types were inferred accordingly.

*6.3. Participation*

Sattler et al. [17] define participation as the willingness to contribute to the development of the organisation. Participation can mitigate scepsis, increase commitment, perceived relevance and identification [56].Participation relates to all stakeholders—professors, students, policy officers, support staff, alumni, future employers, etc. [57]—and it can take many forms: information sessions, queries and polls to gather input, active consultation meetings, expert participation, cooperation and even co-creation. Participation as a strengthening component for a quality culture is more than merely asking for input and consulting stakeholders. It is the willingness to transfer task components, to give stakeholders autonomy and empowerment to shape and execute the component they are responsible for, while maintaining the overall cooperation, focus and alignment of responsibilities [21]. Obstacles for participation and collaboration in universities include the tradition of academic freedom, individualism, competitiveness and identification with the research discipline [1,21]. A strong quality culture is characterized by both a clear alignment of responsibilities and a broad participation in actions and decision making. The leader sets the boundaries and gets out of the way, trusting in self-management and participative decisions. Table 5 lists the types of participation for each of the four archetypes of quality culture.

**Table 5.** Descriptions of the connecting dimensions of the quality culture model.

|  | **Type A** | **Type B** | **Type C** | **Type D** |
|---|---|---|---|---|
| **Leadership** | There is no leader figure or a weak leader. Decisions are rarely made. The decision process takes time and involves conflict. | There is a hierarchical leader who delegates tasks in line with pre-defined structures. The leader makes sure it is clear who is responsible and what results are required. Formal leadership | Leadership is shared and the formal leader acts as a producer. He is task-oriented and defines the expected results. The responsibilities are negotiated and are allocated based on specific interests, skills, experience or resources at that given time. It is a symbiosis where ambitious, energetic staff members who like a specific task are supported to do so by the formal leader. Pragmatic leadership/Strategic leadership/Incremental leadership or Opportunistic leadership. | Leadership is shared and is attributed in an intuitive way. The group members consider the individual and joint responsibilities as self-evident and are happy to contribute. The formal leader acts as facilitator, mentor and stimulates creativity and innovation. Culturally distributed leadership. |

Table 5. *Cont.*

| | Type A | Type B | Type C | Type D |
|---|---|---|---|---|
| **Communication** | There is little information or documentation available. Issues or problems are not discussed. The flow of information between study programme and faculty resp. university is hampered. There is little transparency. There is contradictory information. | The information flow is systematic and selectively tailored to roles. There is formal top-down communication through fixed channels. Bottom-up information is gathered from formal data and queries designed to comply with specific standards. Feedback loops are automated, formal and often partial. | Communication is ad hoc and pragmatic. The availability of information and transparency varies depending on the topic or interest/style of individuals. For some issues there is transparent communication with some or all members. For some issues there is consultation of stakeholders on beforehand and/or dissemination afterwards. | There is a spontaneous consultation and dissemination. Every member asks and gives open, constructive input and feedback to stakeholders. There is maximal transparency and a swift, functional and two-way information flow between all levels of the organization. |
| **Participation** | There is no plan for participation or consultation. There is confusion, it is unclear who to consult. When there is consultation, there is a low to no response. Input is not or partially integrated when taking decisions. Decisions are taken without or with little participation or consultation. | Participation is formally defined and organized by clear procedures and requirements. Jurisdiction and lines of authority are important. There is little or no flexibility in input. Feedback loops are automated and hierarchically organized. | For some tasks there is broad participation and extensive consultation and feedback, often spontaneous. For other tasks the participation is limited to the minimal requirements of a formal procedure or limited to informal ad hoc initiatives. The participation depends on the task, the context or the specific individual. | There is a strong and effective participation along clear task responsibilities. For every task there is an invitation and will to participate in order to obtain the best result. Participation facilitates functional cooperation between stakeholders and integration of task components. |

## 7. Zoom out to Quality Culture

The overall quality culture integrates different aspects of the organisation and can be considered an emergent property. The four archetypes of Table 1 describe a quality culture type at a high, almost abstract level. The descriptions of Tables 3–5 add more detail. Combining Tables 3–5 yields a *maturity matrix* that can be used to identify and structure the variety of quality cultures. The label 'matrix' refers to a structure of reference from which a relief or profile in maturity can surface. For each dynamic component of an existing quality culture, it is possible to relate a specific practice to one of the four archetypical descriptions. The maturity matrix can function as a roadmap to navigate towards a higher level of maturity, inspired by more mature examples of a particular dimension of the quality culture. The maturity matrix can also guide reflection on how the different aspects of a quality culture interrelate and allows the identification of the overall, current level of maturity. As a whole, the model provides a tool to help understand the complex educational context and to help become aware of the different elements that are at play and how these different elements interrelate. When considering the connecting dimensions, the relation with the formal–structural framework is of key importance (Figure 1). This framework has a nested structure (Table 2) and enables the clarification of how every level of the university contributes to educational quality. Within every level of the university these three dimensions of the formal–structural framework should be aligned, providing resources, allowing efficient flows and setting overall clarity and focus. The internal logic should be propagated and should seek feedback through leadership, communication and participation.

Each of the quality culture dimensions can be considered separately, but the main strength of the approach of quality culture is to examine the interaction and interrelation

between dimensions. Indeed, in order to sustain an effective quality culture, the different dimensions should yield an *overall fit*. Coherence of dimensions is necessary for obtaining an effective quality culture [46], regardless of maturity. Strong quality cultures exhibit congruence between policy, ambitions, values and practice. In a weak quality culture, there is incongruence (e.g., strong, hierarchical procedures that conflict with innovation and agility). When all dimensions are oriented towards stability, consistency and control, a Type B formalistic quality culture can be highly effective.

Although the maturity matrix facilitates targeted action, quality culture development demands a systemic approach. Changing only one dimension to a higher level of maturity, e.g., shared values, will not increase the overall effectiveness of the quality culture. On the contrary, in a context where formal–structural frameworks, leadership, participation and human dynamics are geared towards pragmatics and individual initiatives, requiring a wide participation for decision making might be counterproductive. In practice, it will be difficult to change only one dimension, the relations with other dimensions have to be taken into account. Rather than seeing the overall quality culture as the sum of parts, the dimensions can be considered as nodes of a spider web, changing one dimension to a higher maturity level will inevitably affect the others. For instance, a leader who actively communicates their organisation's ambitions and available support and resources can enhance engagement and participation, which in turn feeds into shared values, and a higher level of trust. On the other hand, a defect or obstacle on one dimension can also spread out to other dimensions. For instance, lack of information can lead to uncertainty, which in turn can feed distrust and disengagement. Overall, the model provides a tool for quality development by allowing focus and at the same time drawing attention to the different components of educational practice and the liaisons and interdependencies between them.

In line with Lomas [3] we consider this maturity matrix of quality culture to be a tool *for* quality development—to inform and involve. It is not developed as a tool *of* quality management—to monitor and control. This is for two reasons. First, a self-directed appraisal and growth is more effective for establishing a mature quality culture than a systemic monitoring directive [12,21]. Second, the maturity matrix descriptions have no psychometric properties that warrant diagnostic conclusion. The descriptions are the result of an abductive process, organizing and combining insights from the literature along an overall idea of maturity in quality cultures. The model provides a tool for quality development by identifying different components of educational practice from a birds-eye perspective. The model of quality culture is designed to grasp the layered interplay of formal structures and personal interactions, to encourage ownership and install a perspective of growth. In order to get an idea of its viability, we tested the implementation of the model in a university setting. The following section discusses a pilot study.

## 8. Implementation Test

After designing the fully developed model of quality culture development, we ran a pilot test to answer three questions:

1.   Is the model easy to communicate to educational actors? Do educational actors understand the concept?
2.   Does the model help in structuring the educational context? Does it further the understanding of how educational actors organise their work towards educational quality?
3.   Can educational actors carve out their own current quality profile? Can the model help in setting ambitions for growth?

The specific methodology of the pilot test gives an example of how the model could be used in a university setting. It aims to inspire practices of quality assurance, rather than to be a validation of the theoretical concept model.

*8.1. Setting*

The implementation pilot was run at KU Leuven. KU Leuven assigns every level of the organisation—study programme, faculty and university—the freedom and responsibility to design their own quality assurance, to document the choices made and to assure these choices are fit for purpose. The regulations for quality assurance at the level of the institution are restricted to the minimum, and refer to European standards and guidelines (ESG, [6]) and to criteria that relate to the external accountability of the university. At the university level, supervision and top-down control are light, favouring an open engagement of mutual trust. Given this approach to quality assurance, we believe it makes an interesting case study for testing the implementation of our concept model as an instrument to strengthen quality cultures.

KU Leuven has a central unit that monitors educational quality. The task of this Educational Quality Monitoring Unit (EQM) is to monitor the requirements for external accountability as well as to reach out to every faculty and study programme in order to stimulate and align the local initiatives for educational quality assurance. The EQM unit was assigned the task of communicating, testing and implementing the model. They did not embark on a classic design–approve–implement track but designed the implementation in line with the philosophy of quality culture [9]. Rather than enacting a university-wide policy of implementation, they visited the faculties and study programmes and made them co-owner. By discussing the model, listening to specific practices and stimulating the self-assessment of the different dimensions of quality culture, the implementation process created a two-way learning situation [45].

*8.2. Method*

8.2.1. Participants

The study included four representative faculties: the Faculty of Pharmaceutical Sciences (Biomedical Sciences, small size); the Faculty of Social Sciences (Humanities and Social Sciences, medium size); the Faculty of Science (Exact Sciences, large); and the Faculty of Medicine (Biomedical Sciences, large). Each of these faculties selected two or more study programmes to participate in the pilot study. In total, four faculties and 10 study programmes embarked in the pilot. The interviews at faculty level always included the dean and vice-dean for education but faculties were free to invite others, e.g., administrative directors, programme directors, policy advisors or designated staff members. For the study programme we invited the programme director, who could invite other participants, e.g., programme coordinators, policy advisors or other members of the programme committee (see also: Appendix A).

8.2.2. Procedure

The EQM unit invited each faculty board and study programme for a two-hour interview session. In order to prepare for this interview, texts with an introduction to quality culture, the concept model as well as the full descriptions of the maturity matrix (Tables 3–5) were made available to the participants at least one week prior to the interview. In case clarification or further information was needed, the participants could easily contact the researchers.

There was an elaborate preparation on the side of the EQM unit in order to grasp the specific educational context and to tailor the pre-structured interview to each particular context. Before each interview the EQM unit studied the formal–structural frameworks based on desk-review of websites and several documents. These documents included educational dashboards with the results of student evaluations, a range of study progress indicators (KPI) and self-directed reports on educational quality. We also included their house regulations and regulations for teaching and learning as well as specifics of the local, internal organisation. When additional information was requested, it was readily obtained. Based on their desk-research the EQM unit tailored the interviews in order to enable an informed on-topic discussion of the quality culture.

Each pre-structured interview was conducted by two members of the EQM conducted. The conversation started with a brief introduction and a question-and-answer session on quality culture. The objective of the quality culture monitoring project was made clear: the monitoring project did not imply an evaluation, but a test to jointly map and strengthen their existing quality culture. There was then time for questions from the participants. Consent was asked and each interview was recorded in order to draft and validate the minutes.

The interview was inspired by the appreciative inquiry [58]. The questions were chosen to invite reflection on strengths and opportunities. The questions were non-directive and designed to create a safe setting where there are no 'wrong' answers. For instance, every interview protocol started by asking the participants to give some examples of their good practices. First, the interviews at the faculty level were conducted. At this level the formal–structural dimensions of the quality culture were discussed and reported. There was an interview protocol that focused on the normative, strategic and operative parts of their formal–structural framework. After the interview at the faculty level, we conducted the interviews at the level of the study programme. This allowed us to integrate the description of the formal–structural framework of the faculty in the discussion of quality culture at the level of the study programme. The interview at the programme level started with an invitation to add specifications or adaptations to the formal–structural framework of the faculty in order for this framework to fit to the context of the study programme. Next, the interview at the level of the study programme included three reflection exercises to spark interaction. Then, there were guiding questions that addressed at each of the connecting elements as well as the human dynamics. The participants were asked to assess their own quality culture profile. For each dynamic and connecting dimension of the quality culture they were asked to indicate their current maturity level on a continuum. The continuum was structured by marking four steps: each integer from 1 to 4 corresponded to one of the archetypes from A to D. On each dimension, the integers related to the corresponding description from Tables 3–5. For each dimension, participants were asked to reflect on this continuum, to discuss and then jointly indicate the subjective level of maturity. The participants were asked to motivate their choices as a group. Finally, the participants were asked to carve out the quality profile they wanted to obtain in five years from now. For each dimension, they were asked where they would like to be in five years and to indicate what steps or support they would need in order to work towards this ambition for progress.

The EQM unit drafted a report with the conclusions for each interview. This draft report was then given in ownership to the participants, encouraging them to rewrite, correct or complete what they deemed necessary. They were then to send their final report back to the EQM unit. At the level of the faculty the report contained a description of the formal–structural framework: the education-related objectives; the plans, procedures, tactics and roles put into place to put these objectives into action; and the resources, data and infrastructure made available to support an efficient operationalisation. At the level of the study programme, we documented the adaptations and specifications of their formal–structural framework as documented by their faculty. For each study programme the report documented the connecting and human dynamic elements in the form of two self-reported maturity profiles (current and t+5y). Each profile listed a motivation for the choices made, self-reported steps to take towards growth and the support needed to do so. The EQM included a third maturity profile with their own appraisal of the current maturity level in quality culture.

The initial findings of the pilot were discussed at a university meeting on quality assurance where the different stakeholders—students, professors, and teaching and support staff—could exchange their ideas and reflections. The insights from that discussion are also taken into account.

### 8.3. Findings and Discussion

The limited number of interviews do not allow for far reaching conclusions on how the model is perceived and used. Only tentative conclusions can be drawn. We limit our findings to the observed feasibility and instrumental value of the theoretical concept model for this university setting.

#### 8.3.1. Concept, Model and Maturity

The concept of quality culture was fairly new to the participants, but we experienced an overall positive appreciation of the concept of quality culture. The concept was considered to be congruent with the university's values, policies and practice. Faculties asked to clarify the relation between the mapping of a quality culture, the quality assurance method and other initiatives relating to educational quality.

The radical decision to consider quality culture as a feature of education as a whole, rather than as a description of the system of quality assurance was considered self-evident, especially at the level of the study programmes. At the faculty level there were some questions that referred to the paradigm of measuring educational quality and guaranteeing compliance to standards. Emphasising that the model of quality culture is a tool to understand and strengthen the quality process rather than to measure the quality of the educational output was found helpful.

The overall response to the model was positive. The model was found to be easy to grasp and it triggered interviewees to elaborate on their educational practice and organisation. The interviewees appreciated that the model included the more stable, formal framework as well as the more dynamic, human factors. Leadership, communication and participation were recognised as anchor points for initiating change.

Some faculties were cautious about the idea of maturity. They asked whether it implied a covert top-down ambition, triggering the classic aversion of faculties to comply to university uniformity [9,21]. Further clarification led to a unanimous appreciation of the growth mind-set and an overall striving for excellence in education. Aiming for a quality culture of Type D had full support, without sanctions for quality cultures that are hierarchical or pragmatically organised and with immediate attention to quality cultures with indications of dysfunction (Type A).

#### 8.3.2. Structural Framework and Dynamic Profile

The interviewees were able to relate the model to their own context. The normative, strategic and operative dimensions were found useful to determine and to gain a cognitive hold on the why and the how of the educational organisation. The overview of Table 2 provides an insight into the complex nesting of formal structures that is inherent to a university setting.

The interviewees were convinced that the *formal–structural framework* holds the necessary and enabling conditions for education. However, it was challenging to determine the scope of this framework as well as the appropriate level of detail. In two faculties the reporting was extensive, detailing the normative, strategic and operative dimensions by full description or reference to existing documents. In two other faculties the report was confined to the big picture and listed the specific topics that were discussed during the interview. In the two faculties in which we tested a more elaborate description format we found more resistance than in faculties where we used a more high-level listing and included examples. A first reason for resistance was the work needed to review a fully detailed report with little to no immediate return on that investment. A second reason is that, although the framework is rather stable, there are continuous improvements and changes underway that make these elaborate, full descriptions accurate only for a few years. Although these two faculties were encouraged to design and adapt the prepared documentation of their formal–structural framework, there were many questions, e.g., on how the information would be used by the university. Further consultation is needed to spark a self-directed appreciation of these formal–structural dimensions, to reflect on their

place in the quality culture as a whole, on the relation between educational objectives, strategies and operations, and how these structures translate into the human dynamics. Although this reflective process is complex and time-consuming, it empowers educational actors to understand and fully own their part of the quality assurance process.

The in-depth interviews of the *connecting and human dynamic dimensions* were found to be very pleasant and stimulating for both the interviewees and for members of the EQM unit. Study programmes were intrigued by the group's human dynamic dimensions. These dimensions opened up a rather new perspective on the educational context. For each of the human dynamic and connecting dimensions they indicated their current level on a continuum. The descriptions of Tables 3 and 5 were found to be helpful, enabling them to situate their current practice on the continuum (e.g., between archetype B and C, closer to C than B). Figure 2 gives an example of a quality culture profile from one of the study programmes. In this figure the self-reported ambition to grow (dashed line) is present for all dimensions, yet for some dimensions—shared values or engagement—this ambition is clearer than for others—such as trust or leadership. Although these diagrams are informative, they are not the main goal of the pilot study. We consider the process of exchange and discussion leading to this self-assessment far more valuable. Every mark was motivated and highly contextualised. The annex includes some citations from the interviews, illustrating that the self-assessed level of maturity is far less informative and reliable than the reasons and examples given to motivate their self-assessment. When discussing the connecting dimensions, the interviewees referred to human dynamics such as ownership, shared values and trust as well as to structural elements such as structural procedures, roles, technical processes and infrastructure. For instance, when discussing leadership there was reference to function and task description as part of the strategic dimension. The interview on information often referred to elements of the operative dimensions, e.g., mailing lists, newsletters, websites. Participation referred to strategic dimensions e.g., representation in committees and education-related fora or to operative aspects such as CRS, fixed queries of staff and student evaluation. Overall, the connecting dimensions were indeed recognised as links between a stable, formal framework of education and the human dynamics.

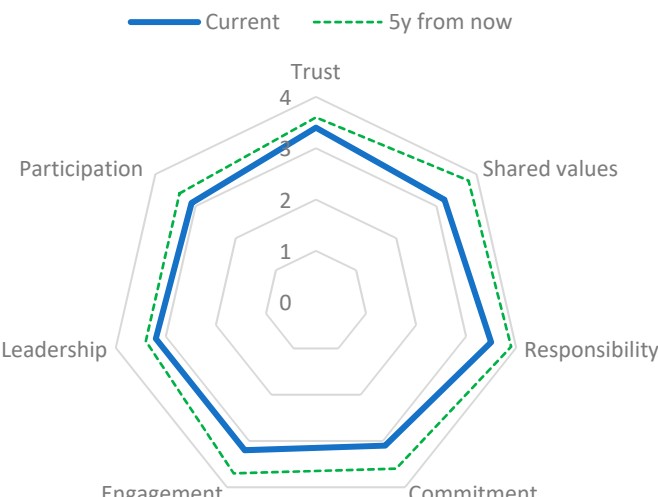

**Figure 2.** Example of a quality culture profile based on self-reported appraisal of connecting and dynamic elements, 1 = Archetype A, 2 = Archetype B, 3 = Archetype C, and 4 = Archetype D.

The resulting quality culture profiles do not pertain to be objective measurements, rather they are snap-shot perceptions and ambitions. They are the result of an interaction and interpretation of reality at a specific moment and should not be considered as a calibrated evaluation. It is possible that educational actors overestimate their current organisational climate, due to bias or other reasons. It is also possible that momentary circumstances lead to an underestimation of the organisational climate, e.g., when there is

a new programme director or during the implementation dip of a big change. In addition, the level of openness and ability for self-critical reflection might covary with the maturity of the quality culture. However, starting a dialogue on educational quality and having actors express their ambitions for growth and further coherence is valuable in itself. The model of quality culture was appreciated in that it provided a structure, an overview, with anchor points to guide the discussion on how their organisation currently functions and how their quality culture could be improved.

### 8.3.3. Approach of Desk-Research, In-Depth Interview and Self-Directed Appraisal

There was an overall appreciation of the quality culture concept, of the perspective of growth and of the insight obtained by self-generating a quality culture profile. Most importantly, reflecting on these initiatives in a face-to-face setting enhanced the confidence of the educational actors and encouraged them to set a clear ambition for growth. Several study programmes expressed a desire to dig deeper into one or more dimensions of the quality culture, e.g., shared values or leadership. They also expressed the ambition to invite other actors for a similar guided discussion on their current quality culture and its maturity. Not only did the EQM unit gain valuable insights on existing practices and issues, appreciating these initiatives stimulated ownership of the quality assurance process. The face-validity of the model of quality culture enables it to be used as a shared mental model. The model of quality culture and the ingrained perspective of growth is promising as a tool for discussing the current way a study programme or faculty functions and how this organisation fosters the educational quality and its assurance.

It should be noted that this interactive approach is time-consuming. It requires substantial effort to desk-research the specific study programmes/faculties, to plan, tailor and conduct the in-depth interviews, to document and co-create the individual profiles on quality culture and to follow-up on reports and requests for support in educational development. We believe that this time-intensive interactive method is needed to establish a baseline for initiating self-directed quality culture development. Once this baseline is established, more sustainable and agile network-based methods of monitoring the quality can be used.

In sum, we found that the concept of quality culture, the model and the idea of maturity are easy to communicate. The overall model and the maturity descriptions provided face-valid markers that enabled educational actors to give examples, to discuss and to self-assess their quality culture. Every faculty could annotate a report of their formal–structural components and all study programmes were able to map and motivate their unique quality profile. They reported that the framework triggered reflection on the overall coherence of underlying dimensions and that it fostered their ambitions to act towards growth.

### 9. Conclusions and Future Steps

Where systems of quality assurance can be mere systems, a strong quality culture fosters education of consistently and sustainable high quality [1,8,9,15,21]. In order to further the understanding of what a quality culture is and how it can be strengthened, we adapted an existing concept model of quality culture [17–19] and enriched it with a perspective of growth [12,23,39]. The resulting model was used as a framework to provide insight how the work on education is organised and how educational actors position themselves. Our concept mapping has two limitations: First, the model of quality culture and the gradations in maturity for each dimension are not designed to have valid or reliable psychometric properties. Rather, the current article focuses on structured cocreation by use of the appreciative inquiry [58]. When setting a safe, appreciative space, the outlined concept model can be used as a tool for structuring and discussing the educational context and for gaining insights into whether and how the different aspects of the organisation of education come together. Second, the model and maturity matrix are constructed based on an abductive research process. Based on literature research we derived definitions and

characteristics for the different model components. Although the model conceptually fits, we did not assess the construct validity or causal pathways. Other approaches did look into the psychometric construct properties of quality culture, e.g., [1,8,17,20], mostly by use of standardised questionnaires. However, for its current purpose—to scaffold the reflection on how one is organised in order to deliver education of high quality—the model appears to be well suited.

The pilot test showed that interacting, discussing and implementing this multi-dimensional model can lay the groundwork for a shared understanding of what defines a quality culture and for an increased insight in how such a quality culture can ensure education of high quality. In our pilot we inferred the feasibility from first-hand process observations. In follow-up research it is advised to include a short survey to assess how interviewees look back on the monitoring of their quality culture. For instance, did the model provide additional insight, did it trigger an ambition to grow, did it further the understanding of quality assurance in education, etc. This information, together with the process observations, can provide converging evidence and allow for a more substantiated answer to the research questions.

Although the model and matrix advanced the subjective understanding and cognitive control on the educational practice, the dimensions have fuzzy boundaries. During the interviews, the three dimensions of ownership—responsibility, engagement and commitment—became intertwined. Additionally, in the literature these concepts are not always interpreted in the same way. It could be interesting to further clarify each dimension and to investigate whether the idea of empowerment can be brought into the picture.

During the pilot test the interviews provided ample time to discuss the overall fit between the different dimensions of quality culture. Although the connecting dimensions linked the formal–structural framework with human dynamics, the reflection on the overall cohesion was not explicitly addressed. It might be interesting to see whether an overall more bureaucratic quality culture (Type B) employs a more elaborate and detailed formal-structural lay-out than a quality culture that is more integrated (Type D). Likewise, a quality culture that is more pragmatically organised (Type C) should have formal structures that are tailored to the degrees of freedom and individual initiatives that are typical for this culture type. In our current approach we merely described the formal–structural dimensions. It can be interesting to include an overall reflection on cohesion. Alternatively, future research can explore whether the formal–structural dimensions can be mapped along the archetypes of quality cultures.

The current project of monitoring the quality culture will be continued throughout the university for all faculties and study programmes. Based on the pilot test there were minor adaptations in the pre-structured in-depth interviews. Next, the human dynamics and connection dimensions at the faculty level will be investigated and integrated with their respective formal–structural frameworks. A similar discussion and mapping should take place at the university level. Although the project of monitoring the quality culture is ongoing, it is clear that this project enhances awareness and ownership in working towards educational quality, marking a key moment for enhancing quality literacy throughout the university [27].

Cultural change is a time-consuming, iterative process requiring intensive commitment [21]. We found an overall positive reception to the model, but this is only partly the result of a comprehensive theoretical forestudy. The model draws its strength from in-the-field interactive discussions with stakeholders. As such our approach is in line with the conclusion of Harvey and Stensaker [10]: *"quality culture first and foremost can be a tool for asking questions about how things work, how institutions function, who they relate to, and how they see themselves. [ . . . ] quality culture is not mechanistic or codified, a system produced by specialists for adoption by others but an iterative, indeed dialectical process of evolution that does not just focus on internal processes but relates to a wider appreciation"* (p. 438). Our approach of strengthening the existing quality cultures can be seen as a collegial, learning experience rather than as a more managerial, policing experience [59]. It is only by continuously

engaging with educational actors that the model has real value, functioning as a powerful tool for zooming in and zooming out on elements of the overall organisation, for reflecting on the work toward excellent teaching and learning, and for appreciating the wonderful intricacy of higher education.

In the last ten years the transition from quality control and assurance to quality culture has made considerable progress. Based on our limited implementation test, we can infer that reaching out, discussing and mapping out quality cultures can function as key moments for enhancing awareness, setting out a shared mental model and appreciation of quality culture. Although the paradigm of quality control still lingers, we gradually found more confidence in, and support for, the idea of quality culture. The paradigm shift is still ongoing, and time is needed to further implement and sustain this transition to quality culture. Continuing dialogue and interaction with and among individual stakeholders and policy units will strengthen the impact of the quality culture approach as a ground for fostering an intrinsically motivated quality assurance in higher education.

**Author Contributions:** Conceptualization N.V. and J.V.D.; methodology N.V., J.V.D., A.V. and Y.S.; formal analysis N.V. and Y.S.; investigation, J.V.D. and N.V.; resources, A.V. and M.B.; data curation, Y.S.; visualisation: N.V. and Y.S.; writing—original draft preparation, N.V. and J.V.D.; writing—review and editing, N.V.; supervision, A.V. and M.B.; project administration, Y.S. All authors have read and agreed to the published version of the manuscript.

**Funding:** This research received no external funding. This study is part of the larger project of monitoring the quality culture at KU Leuven.

**Institutional Review Board Statement:** The study was conducted in accordance with the Declaration of Helsinki and received feedback of the Ethics Committee SMEC - Sociaal-maatschappelijke Ethische Commissie of KU Leuven (protocol code G-2022-5547; dd. 15/12/2022).

**Informed Consent Statement:** Informed verbal consent was obtained from all subjects.

**Data Availability Statement:** Data is contained within the article.

**Acknowledgments:** We thank Celine De Vos and Nele Annaert for their contributions to the theoretical concept. We thank the members of the Werkgroep Onderwijskwaliteit KU Leuven for their valuable feedback on the pilot study.

**Conflicts of Interest:** The authors declare no conflict of interest.

## Appendix A  Examples of Some Motivations Interviews Gave When Finding Their Own Level of Maturity, as Guided by the Descriptions of Archetypes

PEC refers to the (permanent) education committee overseeing the study program. In this committee all stakeholders are represented—professors, students, alumni and staff (administrative, student counselling, etc).
Responsibility:

> "There is a feeling of responsibility, expectations are met and when prompted by the programme director, we also see some self-directed responsibility" (Type B to C)

> "Responsibility is situated at different levels. Growth towards more support and more creativity should be possible. For this, people need time and resources. People also take responsibility only when they can benefit from it. Now it is more about a calculated commitment." (Type C to D)

Engagement:

> "Although the majority of agenda items have a formal/expressed interest. There is sometimes enthusiasm, but also apathy or opposition. Overall, I feel that PEC members are less likely to volunteer for tasks." (Type B)

"Because of the high workload and broad assignments (teaching/research/public service/clinic duty) of professors, a continuously high engagement is not possible." (Type C)

Commitment:

"Individual lecturers are mainly focused on and committed to assuring the quality of their courses. Where there is room for improvement: the commitment to the programme as a whole and involvement to contribute to the whole study programme." (mid-Type B and C)

Trust:

"We favour an open consultation culture. However, in the PEC, more fundamental issues are sometimes discussed in hedged, covert terms, to avoid targeting specific persons. This is often discussed in advance with the person(s) involved. The direct colleagues [professors] are all member of the same research unit, making it easy to review and discuss issues in advance. We feel that trust among students is high, meaning that they (can) raise topics at the PEC." (Type C)

"The members of the PEC changed little in the past 10 years. We know each other very well; we understand each other and easily decide on who does what. Because of this high mutual trust, cooperation within the PEC runs smoothly. Because of the high level of trust in our programme director, the leader is in charge of the day-to-day operations. He works out a proposal and asks the members for approval." (Type D)

Shared values:

"The PEC takes a pragmatic approach. The study programme is legally bound to specific content to be covered. The consultations are mainly virtual/online, unless the topic requires physical consultation. Many members of the PEC also work at the university hospital. Because of the agendas, online consultation is preferred. So there is somewhat less need for continuous development, we are trying to work efficiently. The focus is mainly to reach the targets in a most efficient way." (Type C)

Leadership:

"I feel that level C fits here—there is a soft steering with ample space for task assignment and responsibility." (Type C)

"We see the role of programme director as a responsibility, not an authoritarian position. The PEC is prepared in advance but at the PEC there is an opportunity to give feedback on the agenda or to introduce new agenda items. Decisions are made through dialogue and consensus; no formal voting is used." (Type C)

"[As a programme director] I take on a mediating, coaching role—taking responsibility within the programme, but based on the input, vision and critical reflections of the PEC members and after a consensus has been reached. I find it very important to show gratitude to the members of the PEC, appreciating their work, [ ... ] the programme coordinator puts slightly more focus on formal rules and policies. We both find it important that rules and agreements are clearly communicated and respected, both towards students and teachers." (Type D—maximum)

Communication:

"There is an open communication culture in the PEC and among its members (also outside the PEC). This is the advantage of a rather small-scale PEC and the fact that there is no competition among teachers. There is spontaneous consultation and feedback." (Type D)

"Various channels (website, SharePoint) are used to get information to stakeholders, but the programme director has no idea on whether this info is consulted. Good practices are shared at the PEC. Sometimes the programme director uses

email for this, but also telephone or face-to-face contact. As it is a small course, with a small group of lecturers, contacts are more informal. Certain elements are discussed in advance with the lecturer, after which they are referred to in the PEC." (Type C)

Participation:

"The PEC takes a pragmatic approach. The consultations are mainly electronic, unless the topic requires physical consultation. Many members of the PEC also work at the university hospital. Because of the agendas, electronic consultation is preferred." (Type C)

"Teachers of the programme get involved through a teachers' day (formal) and through informal contacts in the research department. There is a representation of teachers present in the PEC. There are student representatives in the PEC. Students can bring their agenda items to the PEC and can discuss them before the meeting." (Type B and C)

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
