# Peer review of "A Maturity Matrix Model to Strengthen the Quality Cultures in Higher Education"

_education, doi:10.3390/educsci13020123_

Round 1

Reviewer 1 Report

The issue of quality culture research is not new, but still relevant and important for the development of universities. The article is interesting and can make a significant contribution to literature and practice. Methods and tools are still needed to help diagnose the quality culture not only globally, but also in relation to smaller groups. The paper is clear, appropriate for a journal, and well organized. The conducted analysis and the way of presenting the research results are correct. However, the article uses the most recent literature to a small extent (within the last 5 years).

In my opinion, some improvements are possible which will increase the value and attractiveness of the paper. In the introduction, I suggest that you improve the presentation of the aim of this study. It is important that you clearly state the novelty of your study and the contribution it can make to the literature. Currently, this is not clearly addressed. There are tools in the literature for measuring quality culture, not only in general terms, but also in the context of applying various qualitative concepts in universities (for example, the Lean Culture Maturity Model in HE). The article would be more complete if it indicated what distinguishes the proposed method from others in relation to universities. I propose to move the description of the literature search method from point 3.2 to point 3, as it relates to the content of both points 3.2 and 3.3. In section 3.2 it is described that the search terms used in conjunction with "quality culture" and "higher education" were "leadership, communication, information, shared values, trust, responsibility, commitment or engagement". This set does not coincide with the model elements listed in Figure 2. Instead of "information" there should be "participation". In point 4.2.2. it is worth clarifying the description of the focus study. What was his purpose? What exactly was discussed?

I evaluate the paper positively. I believe that it may interest readers and contribute to the popularization of research on quality culture in higher education.

Reviewer 2 Report

The paper returns to the important on-going issue of how to develop an effective quality culture in higher education.  The development of the discussion of maturity in quality culture is particularly interesting. As such, it is of continuing interest and there is much here to commend the paper and the research behind it.  However, there are a number of issues that would need addressing before it could be published.

The initial concern relates to clarity, which is, in part structural and in part the approach you have taken.  I don't feel that you are clear from the start about what you actually did - you state this in the abstract, that you did a review of the literature and a pilot study, but not really very much about it until much later on in the main text.  I would argue that perhaps the focus should be the pilot study (with the literature justifying and informing this piece of research).  Perhaps the problem in it current form is that you are trying to do too much?

The literature review is quite descriptive and does not always form part of a clear, tight argument.   The discussion, in places, is determined by selected authors rather than summarising and discussing key themes and issues.  It is all there, but perhaps work is needed to refine and reduce this.  The seemingly obsessive concern with Sattler and Sontag is a little disconcerting when you mention a lot of other scholars in the field, who, quite frankly, are leading the discussion (eg. Harvey and Stensaker, who are not listed in your references, Ming Cheng). I would suggest you engage more with Lee Harvey's work, which is a valuable resource (whether you agree with him or not!) - for example, he published a piece, relatively recently (2017) on quality culture.

I would like to see more justification/ rationale of the case study of KE Leuven. Currently, it appears to be chosen rather randomly or that this is an institutional report that has been turned into an article.

Your pilot study refers to interviews but you don't quote anyone: you are not letting your participants speak for themselves. This would be interesting data.

For me, the use of the standard structure does not really help your argument.  The separation between findings and discussion leads to an unnecessary split between your evidence, which is qualitative, and discussion.  This takes away from the overall argument.

I was a bit confused by the figures.  Figure 1 contains useful elements but I feel that the items at the top: study programme, faculty and university seem to be unconnected. Surely these are heavily interconnected?  Surely, the sections within 'quality culture' should be interconnected, too?  I also feel that you can't set study programme, faculty and university outside quality culture - QC should cover everything and the figure implies separation! Figure 2 is pretty unreadable and I wonder if it could be simplified?

Although they have their place, I am not generally keen on bullet points in articles.  In this case, the bullet-points make the whole paper seem like a list of things or almost like a text book of things for students to remember, rather than a flowing argument from start to finish.

The use of the standard structure does not really help your argument.  The separation between findings and discussion leads to an unnecessary split between your evidence, which is qualitative, and discussion.  This takes away from the overall argument.

I think you need to be clearer about what you actually mean by a 'maturity matrix'.  I found it difficult to find any clear discussion of this.  Yet it is a central platform of your paper.  This is a common issue in research on higher education: such phrases and concepts need clearer description!

Overall, then, I think the paper has something interesting to say and I would like to see it published eventually.  However, it needs further refinement and clarification along lines suggested to make into a tight, well argued paper.  

Round 2

Reviewer 2 Report

The article is very much better and in my view could now go forward.  One minor point - the Barnett reference is Barnett, R. (Ronald), not Barnett, D.!

Author Response

Many thanks for the positive reaction. 
I have corrected the initial of Barnett in the references, thanks for pointing that out. 

Kind regards